# Butyl Methacrylate-Co-Ethylene Glycol Dimethacrylate Monolith for Online in-Tube SPME-UHPLC-MS/MS to Determine Chlopromazine, Clozapine, Quetiapine, Olanzapine, and Their Metabolites in Plasma Samples

**DOI:** 10.3390/molecules24020310

**Published:** 2019-01-16

**Authors:** Luiz G. M. Beloti, Luis F. C. Miranda, Maria Eugênia C. Queiroz

**Affiliations:** Departamento de Química, Faculdade de Filosofia Ciências e Letras de Ribeirão Preto, Universidade de São Paulo, 14040-901 Ribeirão Preto, SP, Brazil; lgmbeloti@usp.br (L.G.M.B.); luisfelipe.c22@usp.br (L.F.C.M.)

**Keywords:** in-tube SPME, UHPLC-MS/MS, organic-based monoliths, antipsychotics, plasma samples, schizophrenic’ patients

## Abstract

This manuscript describes a sensitive, selective, and online in-tube solid-phase microextraction coupled with an ultrahigh performance liquid chromatography-tandem mass spectrometry (in-tube SPME-UHPLC-MS/MS) method to determine chlopromazine, clozapine, quetiapine, olanzapine, and their metabolites in plasma samples from schizophrenic patients. Organic poly(butyl methacrylate-co-ethylene glycol dimethacrylate) monolith was synthesized on the internal surface of a fused silica capillary (covalent bonds) for in-tube SPME. Analyte extraction and analysis was conducted by connecting the monolithic capillary to an UHPLC-MS/MS system. The monolith was characterized by scanning electron microscopy (SEM) and Fourier transform infrared spectrometry (FTIR). The developed method presented adequate linearity for all the target antipsychotics: R^2^ was higher than 0.9975, lack-of-fit ranged from 0.115 to 0.955, precision had variation coefficients lower than 14.2%, and accuracy had relative standard error values ranging from −13.5% to 14.6%, with the exception of the lower limit of quantification (LLOQ). The LLOQ values in plasma samples were 10 ng mL^−1^ for all analytes. The developed method was successfully applied to determine antipsychotics and their metabolites in plasma samples from schizophrenic patients.

## 1. Introduction

Schizophrenia is a severe and chronic mental disorder characterized by profound disruptions in thinking, which consequently affects language, perception, and the sense of self [1]. This disorder is characterized by positive (psychotic behaviors), negative (disruptions to normal emotions and behaviors), and cognitive symptoms (changes in memory or other aspects of thinking) [1]. Atypical antipsychotics are the mainstay of treatment prescribed to schizophrenic patients. Compared to classic neuroleptics, these drugs induce fewer extrapyramidal syndromes [1,2].

Studies have suggested that the pharmacokinetics of atypical antipsychotics involve large inter- and intra-individual differences among patients (age, gender, lifestyle, genetic and metabolic characteristics, and drug interactions). Therefore, therapeutic drug monitoring (TDM) can be extremely useful to establish an effective individual therapeutic dose that maintains plasma drug concentrations within a targeted therapeutic range, thereby avoiding an overdose [3,4].

Liquid chromatography–tandem mass spectrometry (LC-MS/MS) is a highly sensitive and selective technique to analyze drugs in biological fluids quantitatively. Biological fluids are complex matrixes containing endogenous macromolecules that can irreversibly adsorb on the analytical column stationary phase surface, which reduces chromatographic separation efficiency. Moreover, during MS/MS analysis, nonvolatile solutes can suppress ionization (electrospray ionization), which decreases the LC-MS/MS method sensitivity [5]. These difficulties call for sample preparation to diminish the matrix effect. This step increases not only the sensitivity, but also the selectivity of the LC-MS/MS method. 

In-tube solid-phase microextraction (in-tube SPME) is a sample preparation technique that uses a capillary column, as extraction device, directly coupled to a LC system. The in-tube SPME-LC system is fast to operate, easy to automate, and environmentally friendly (organic solvent is only used in the mobile phase). Automated methods always provide better accuracy and precision than manual procedures [5,6,7]. Capillary columns with different selective stationary phases (coating), including restricted access media (RAM), molecularly imprinted polymers (MIP), immunosorbents, and monolithic materials, have been used for in-tube SPME systems [8,9,10,11,12]. 

In-tube SPME-LC methods with different organic monolithic capillaries have been applied for analysis of several analytes. For example: poly (methacrylic acid–ethylene glycol dimethacrylate) for basic drugs in human serum [13], and amphetamines in urine [14], NH_2_-MIL-53(Al) incorporated poly(styrene-divinylbenzene-methacrylic acid) (poly(St-DVB-MAA)) for estrogens in human urine [15], and (*N*-isopropylacrylamide-co-ethylene dimethacrylate) for acid, basic, and neutral compounds [16]. Monolith materials with good control of porosity, diverse surface chemistry, and frit-free are easy to prepare by in situ polymerization [15,16,17]. The monolithic porous structure facilitates convective mass transfer (which is preferable during extraction) with reasonably low pressure. Organic-based monoliths are stable within the entire pH range and biocompatible with biological samples [17,18].

This manuscript describes an in-tube SPME-UHPLC-MS/MS with an organic poly(butyl methacrylate-co-ethylene glycol dimethacrylate) monolithic capillary to determine antipsychotics (chlopromazine, clozapine, quetiapine, and olanzapine) and their metabolites (desmethyl chlorpromazine, 7-hydroxy-chlorpromazine, *N*-desmethyl clozapine, *N*-desmethyl olanzapine, and norquetiapine) in plasma samples from schizophrenic patient. Figure 1 illustrates the metabolic and biotransformation pathways of these antipsychotics.

## 2. Results and Discussion

### 2.1. Organic Poly(Butyl Methacrylate-Co-Ethylene Glycol Dimethacrylate) Monolith Capillary Preparation and Characterization

Although the proposed monolith synthesis was based on classical radicalar procedures [19,20], the innovation of this work is related to direct coupling of the poly (butyl methacrylate-co-ethylene glycol dimethacrylate monolith capillary (in-tube SPME) to the LC-MS/MS system. 

The cross-linking monomer (type and crosslink density), the porogenic solvent (type and amount), and the ratio between the functional and the cross-linking monomers substantially influence the polymer surface area, pore volume, pore size, and porosity [17]. Pore size distribution must be adjusted during monolith preparation, so that the monolith fits the desired application [18].

Table 1 illustrates optimization of the conditions of the organic poly(butyl methacrylate-co-ethylene glycol dimethacrylate) monolith capillary synthesis procedure. The porogenic solvent controls the organic monolith porosity [21,22]. The porogenic solvents 1-propanol and 1,4-butenodiol employed here produced a homogenous pre-polymer solution containing the monomers. A slight increase in the amount of porogenic solvent generated larger pores, which improved monolithic capillary permeability and favored sample percolation through the capillary under the in-tube SPME system low pressure.

Alteration in the cross-linking monomer percentage in relation to the functional monomer modified monolith porosity. High cross-linking monomer concentration decreased both pore size and permeability.

On the basis of analyte extraction efficiency (Figure 2), the functional and cross-linking monomer percentages were optimized (Table 1). The M3 monolithic capillary (Figure 2) was selected for the in-tube SPME-LC analysis because it gave the highest extraction efficiency and adequate permeability.

Figure 3 shows the cross-sectional SEM images of poly(butyl methacrylate-co-ethylene glycol dimethacrylate) monolith capillary at 200× and 10,000× magnification. The monolith exhibited continuous (greater homogeneity) coating with interconnected macropores, which allowed the sample to be percolated through the capillary at low pressure. The monolith was clearly tightly attached to the capillary inner-wall.

The monolith was fixed to the inner capillary surface by covalent bonds, which dismissed the need for frits [22]. As a result of this chemical attachment and its structure, the monolith exhibited mechanical and chemical stability, as well as biocompatibility with biological samples [22,23]. The proposed monolithic capillary was reused more than one hundred times without observing significant changes in sorption capacity.

Figure 4 depicts the monolithic phase FTIR spectra. The bands at 2960.7 and 1454.4 cm^−1^ indicated bond between carbon sp^3^ and hydrogen atoms. The band at 1728.6 cm^−1^ corresponded to carbonyl bond. The bands at 1388.9, 1254, and 1157.1 cm^−1^ were ascribed to symmetric and asymmetric ester C-O bond vibrations. The spectra also displayed bands at 1637.6 cm^−1^, due to residual vinyl C=C bond stretching; at 814.36 and 751.24 cm^−1^, attributed to cis-substituted vinyl group vibration; and at 959.12 cm^−1^, assigned to C-H bond out-of-plane vibration. The band at 3443.8 cm^−1^ referred to adsorbed water hydroxyl groups, a consequence of the monolith phase high porosity. The FTIR spectra confirmed incorporation of both monomers, butyl methacrylate and ethylene glycol dimethacrylate, in the monolithic capillary structure [24,25].

### 2.2. Plasma Sample Pre-Treatment

After protein precipitation, the dried extract was reconstituted with 100 µL of 5 mM ammonium acetate solution at different pH values. At pH 10, analytes were in non-ionized or the partially ionized form, which improved hydrophobic interactions between the analytes and the monolithic capillary.

As reported in the literature, the poly(butyl methacrylate-co-ethylene glycol dimethacrylate) monolith is selective for hydrophobic analytes, such as polycyclic aromatic hydrocarbons in smoked meat products [19] and cyclophosphamide and busulfan in whole blood samples [20].

The pre-treatment step boosted extraction efficiency because it attenuated the matrix effect.

### 2.3. Analytical Validation

Analytes were detected by ESI–MS/MS in the SRM and in the positive ionization modes. The protonated molecules of the analytes [M + H]^+^ were fragmented by collision-induced dissociation (CID). Two product ions of each analyte were selected as transitions in the SRM detection mode: one for quantitative and the other for qualitative purposes (Table 2).

Method selectivity was assessed by comparing a blank plasma sample chromatogram with the chromatogram of a blank plasma sample spiked with the target analytes at concentrations corresponding to the lower limit of quantification (LLOQ) (Figure 5).

The developed method was linear from the LLOQ (10 ng mL^−1^) to the upper limit of quantification (ULOQ) (200 ng mL^−1^ to 700 ng mL^−1^); the coefficient of determination was higher than 0.9975 (Table 3). The lack-of-fit test confirmed method linearity. Calibration standards (*n* = 5) presented coefficient of variation (CV%) lower than 15%. This linear range included therapeutic intervals.

The developed method presented accuracy, with RSE values ranging from −19.4 to 19.9% (intra-assay) and from −18.9 to 19.3% (inter-assay), as well as precision, with CV values ranging from 0.7 to 14.2 (intra- and inter-assay) (Table 4).

The method matrix effect was evaluated by using CV values of the IS-normalized matrix effect that were lower than 15% (Table 4). Residual carryover in blank plasma samples following ULOQ analysis exhibited values lower than 3% of the analyte LLOQ signal, or 0.05% of the IS LLOQ signal.

Comparing the in-tube SPME-UHPLC method with literature methods (Table 5), the proposed method presented lower LLOQ values than the values obtained with the MEPS-UHPLC [26] method. The proposed method presented the lowest chromatographic separation time (Table 5). Moreover, the proposed method used reduced plasma sample volume (300 µL) and provided simultaneous determination of different antipsychotics and their metabolites. 

### 2.4. Determination of Antipsychotics and Their Metabolites in Plasma Samples from Schizophrenic Patients

The proposed method was successfully applied to determine the target antipsychotics in plasma samples from three schizophrenic patients undergoing therapy with atypical antipsychotics (Table 6). Consequently, this method could be used for therapeutic drug monitoring.

## 3. Materials and Methods 

### 3.1. Standards and Reagents

Chlorpromazine, clozapine, olanzapine, and quetiapine were purchased from Cerilliant (Round Rock, TX, USA). Desmethylchlorpromazine and 7-hydroxy-chlorpromazine were acquired from TRC Canada (Toronto, ON, Canada). *N*-desmethylclozapine was obtained from Sigma-Aldrich (St. Louis, MO, USA). *N*-desmethylolanzapine was purchased from Santa Cruz Biotechnology (Dallas, TX, USA). Norquetiapine was acquired from Biovision (Milpitas, CA, USA). The internal standards chlorpromazine-d3 and quetiapine-d8 were obtained from Cerilliant (Round Rock, TX, USA). Butyl methacrylate (BMA) (99%), ethyleneglicol dimetacrylate (EGDMA, 98%), 1-propanol (HPLC grade), vinyltrimethoxysilane (VTMS) 1,4-butendiol (99%), and 2,2-azobisisobutylnitrile (AIBN) were purchased from Sigma–Aldrich (St. Louis, MO, USA). Fused silica capillary (530 μm i.d. × 10 cm) was acquired from NST (São Paulo, Brazil). Acetonitrile, methanol (HPLC grade), ammonium acetate, and formic acid were supplied by JTBaker (Phillisburg, NJ, USA). Hydrochloric acid and sodium hydroxide were purchased from Sigma–Aldrich (St. Louis, MO, USA). Aqueous solutions were prepared with ultrapure water from a Milli-Q, Millipore system (18.2 MΩ cm) (São Paulo, SP, Brazil).

### 3.2. Organic Poly(Butyl Methacrylate-Co-Ethylene Glycol Dimethacrylate) Monolith Capillary Preparation

The monolith was synthesized based on published literature [19,20], with modifications. The fused silica capillary was pretreated to activate surface silanol groups. The capillary was initially rinsed with 0.2 mol L^−1^ HCl for 30 min, which was followed by water until the outlet solution achieved pH 7.0. Subsequently, the capillary was flushed with 1 mol L^−1^ NaOH for 2 h, which was followed by water and methanol for 30 min. Finally, the capillary was purged with nitrogen at 160 °C for 3 h prior to use. To achieve covalent binding between the polymer materials and the capillary inner wall, the capillary was modified with vinyltrimethoxysilane solution. The activated capillary was then silanized as previously reported by Ho, T.D. et al. [29]. The capillary was filled with VTMS, sealed with silicon rubbers, and reacted at 85 °C for 2 h. The silylated capillary was rinsed with MeOH and purged with nitrogen at 60 °C in a GC oven for 3 h, to give the vinyl-functionalized capillary.

To perform polymerization, different BMA (functional monomer), EGDMA (cross-linking monomer), AIBN (radicalar initiator), and 1-propanol and 1,4-butenodiol (porogenic solvents) proportions (Table 1) were mixed (vortex for 1 min), sonicated for 10 min, and purged with a nitrogen stream for 10 min. The activated capillary was filled with this mixture, and both capillary ends were sealed with silicon rubbers. The polymerization reaction was kept at 60 °C for 20 h. Finally, the capillary was rinsed with methanol for 2 h to remove unreacted monomers, porogens, and any other soluble compounds from the pores.

### 3.3. Organic Poly(Butyl Methacrylate-Co-Ethylene Glycol Dimethacrylate) Monolith Capillary Characterization

Scanning Electron Microscopy was employed to evaluate organic monolith surface morphology. Samples were submitted to carbon evaporation and were coated with gold for 180 s in a Bal-Tec SCD050 Sputter (Fürstentum, Liechtenstein). The samples were then analyzed in a Carl Zeiss EVO5O scanning electron microscope (Cambridge, UK). The chemical groups present on the monolith were identified by Fourier Transform Infrared Spectroscopy (FTIR) on the Shimadzu-IRPrestige-21 (ABB Bomem series MB100) spectrometer; KBr pellets were used.

### 3.4. Plasma Samples

The developed in-tube SPME-UHPLC-MS/MS method was optimized and validated with drug-free plasma (blank samples) from volunteers that had not been exposed to any drug for at least 72 h. These blank plasma samples and plasma samples from patients undergoing therapy with antipsychotics were kindly supplied by the Psychiatric Nursing staff of the University Hospital of Ribeirão Preto Medical School, University of São Paulo, Brazil. The plasma samples were collected in agreement with the criteria established by the Ethics Committee of Ribeirão Preto Medical School, University of São Paulo, Brazil. The plasma samples from schizophrenic patients were collected and stored at −80 °C for six months.

### 3.5. Plasma Sample Pre-Treatment

Initially, plasma sample (300 µL) was spiked with internal standard solutions (Chlorpromazine-d3 and Quetiapine-d8 at 50 ng mL^−1^). Plasma proteins were precipitated with acetonitrile (600 µL), and then after centrifugation for 20 min, 800 µL of the supernatant was collected and dried in the vacuum concentrator (Eppendorf, Brazil). The dried extract was reconstituted with ammonium acetate/ammonium hydroxide solution (5 mmol L^−1^). Considering the monolith sorption capacity, different pH values (4.0, 7.0, and 10.0) of this solution were evaluated. Then, 10 µL of this solution was injected into the in-tube SPME-UHPLC-MS/MS system.

### 3.6. LC-MS/MS Conditions

LC–MS/MS analyses were performed on a Waters ACQUITY UPLC H-Class system coupled to the Xevo^®^ TQ-D tandem quadrupole mass spectrometrer (Waters Corporation, Milford, MA, USA); a Z-spray source operating in the positive mode was used. The optimum parameters were: capillary voltage of 3.20 kV, source temperature of 150 °C, desolvation temperature of 400 °C, desolvation gas flow of 700 L h^−1^ (N_2_ 99.9% purity), and cone gas flow of 150 L h^−1^ (N_2_ 99,9% purity). Analytes were analyzed in the selected reaction monitoring (SRM) mode. Argon (99.9999% purity) was employed as collision gas, and the dwell time for each transition was set to 0.049 seconds. The analytes were separated on an ACQUITY UPLC CSH C18 (1.7 µm, 2.1 × 100mm) column at 40 °C, and data were acquired by using the MassLynx V4.1 Software (Waters Corporation, Milford, MA, USA).

### 3.7. In-Tube SPME-UHPLC-MS/MS Configuration

Two columns were connected by means of a six-port valve, as shown in Figure 6. The monolithic column (first dimension) was connected to the quaternary pump (QSM), and the analytical column (second dimension) was connected to the binary pump (BSM). In the first step, the six-port valve was set in position 1, which allowed the columns to be conditioned with the initial mobile phase composition (Table 7). Then, 10 µL of the sample solution was injected into the system. Water was used as mobile phase to percolate the sample solution through the monolithic capillary at a flow rate of 100 µL min^−1^. In this step, analytes were pre-concentrated, and macromolecules from plasma samples were eluted for waste. After 2.0 min, the six-port valve was set to position 2. The target drugs from the monolithic capillary were eluted to the analytical column using the mobile phase (BSM pump), consisted of (A) 10 mmol L^−1^ ammonium acetate (with 0.1% formic acid) and (B) acetonitrile (80:20 *v*/*v*), at a flow rate of 100 µL min^−1^. At 6.50 min, the six-port valve was set to position 1. Using the same mobile phase, from 6.51 to 13.00 min, the chromatographic separation occurred at 300 µL min^−1^. From 7.0 to 16.5 min, acetonitrile was percolated through the monolithic capillary for clean-up. After this time, both columns were conditioned with the initial mobile phase composition for the following sample injection (Table 7). Different mobile phases were evaluated to establish the highest analyte sorption and the best analyte resolution within the minimum analysis time.

### 3.8. Analytical Validation

Based on current international guidelines of the European Medicines Agency (EMA) and Food and Drug Administration (FDA), the in-tube SPME-LC-MS/MS method was validated.

Using linear regression of the ratio between the peak area of the target drugs and the IS (Y-axis) peak area versus the nominal drug concentrations (X-axis, ng mL^−1^), the calibration curves were generated.

The lower limit of quantitation (LLOQ) is the lowest concentration in the calibration curve that can be quantitatively determined with suitable precision and accuracy.

Accuracy was evaluated from relative squared error (RSE) values. Precision was estimated from the CV values of the analyses of the blank plasma samples spiked with drugs at five different concentrations (*n* = 5), namely LLOQ, low quality control (QC), medium QC, high QC, and upper limit of quantitation (ULOQ).

Matrix effects were investigated by using eight lots of blank plasma obtained from different sources spiked with drugs at low QC and high QC concentrations. The matrix factor (MF) was evaluated for each matrix lot, by comparing the drugs responses in the presence of matrix with those in the absence of matrix. The IS normalized MF was also calculated by dividing the drug MF by the IS MF. The CV of the IS-normalized MF calculated from the eight matrix sources should not exceed 15%. 

The carry-over should be evaluated by analyzing a blank sample following the highest concentration calibration standard. The response in the blank sample obtained after measurement of the highest concentration standard should not be greater than 20% of the analyte response at the LLOQ and 5% of the response of internal standard.

## 4. Conclusions

The organic poly(butyl methacrylate-co-ethylene glycol dimethacrylate) monolith developed herein exhibited low backpressure, high permeability, and adequate sorption to determine antipsychotics and their metabolites in plasma samples at sub-therapeutic levels.

Compared to other microextracion techniques, the automate in-tube SPME system allows direct coupling of the microextraction step to chromatographic systems, which not only increases the accuracy and precision, but also reduces the organic solvent consumption and analysis time.

The in-tube SPME-UHPLC-MS/MS method exhibited good selectivity and sensitivity due to analyte pre-concentration in monolithic capillary. This method was successfully applied to determine antipsychotics and their metabolites in plasma samples from schizophrenic patients.

## Figures and Tables

**Figure 1 molecules-24-00310-f001:**
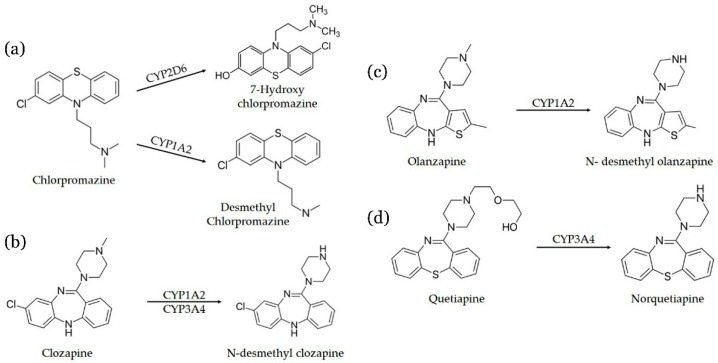
Biotransformation pathways of (**a**) Chlorpromazine, (**b**) Clozapine, (**c**) Olanzapine, and (**d**) Quetiapine. CYP = cytochrome P450 complex. CYP1A2, CYP2D6, and CYP3A4 are isoenzymes of cytochrome P450 complex.

**Figure 2 molecules-24-00310-f002:**
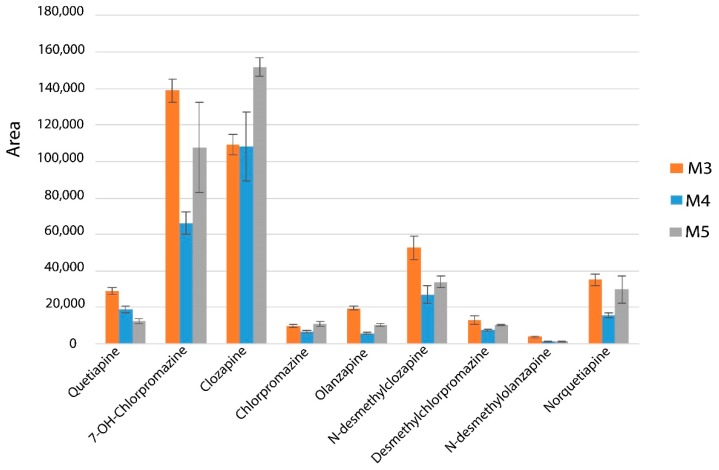
The effect of different synthesis procedures on the performance of in-tube SPME-MS/MS method (Table 1 describes the synthesis conditions).

**Figure 3 molecules-24-00310-f003:**
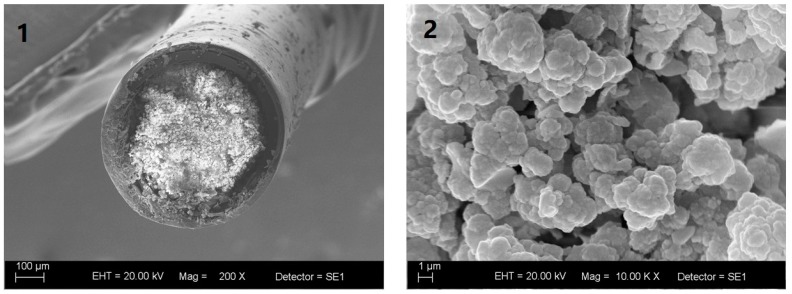
Scanning electron microscope images of the cross-section of poly(butyl methacrylate-co-ethyleneglicol dimethacrylate) monolith capillary at (**1**) 200× magnification and (**2**) 10,000× magnification.

**Figure 4 molecules-24-00310-f004:**
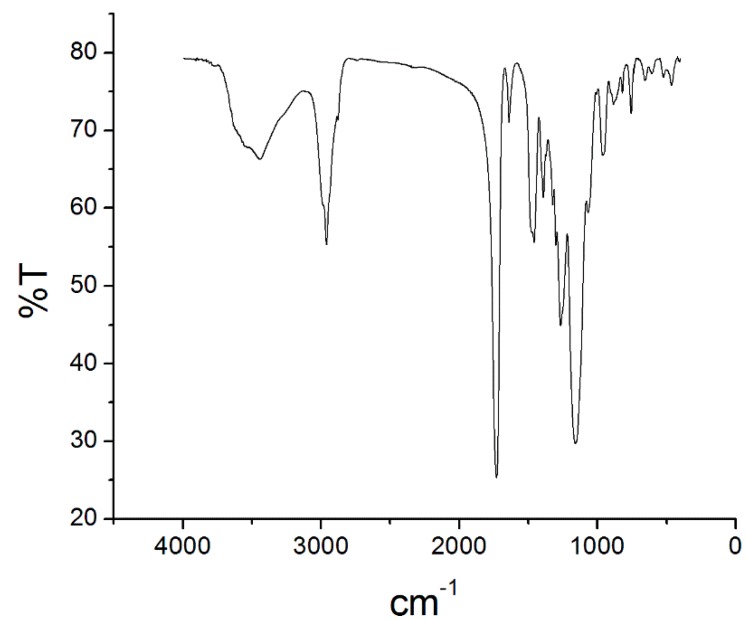
Fourier transform infrared spectrometry spectra of the poly(butyl methacrylate-co-ethyleneglicol dimethacrylate) monolith capillary.

**Figure 5 molecules-24-00310-f005:**
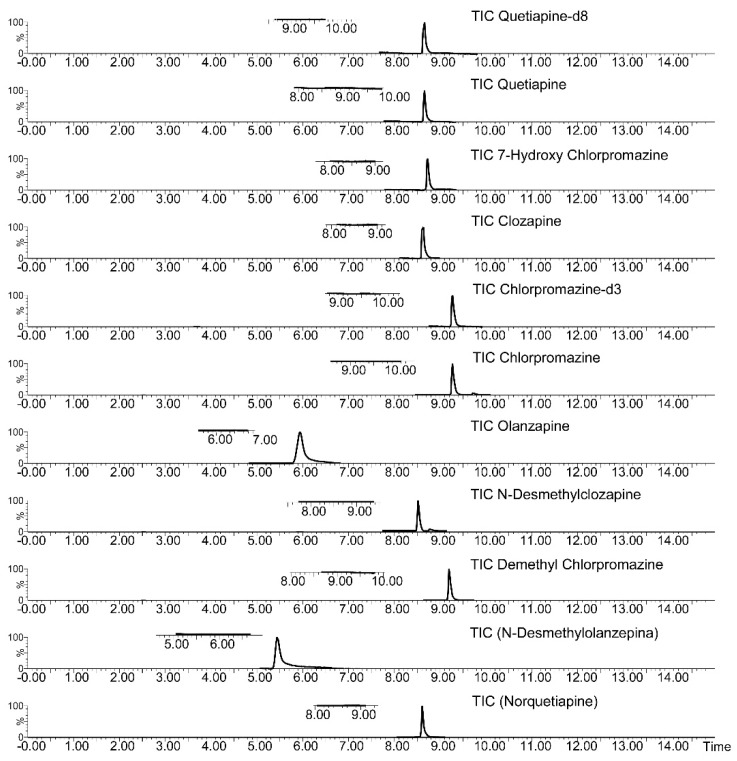
LC-MS/MS TIC (Total Ions Current) chromatograms of a drug-free plasma sample (subscript on the left) and drug-free plasma sample spiked with target drugs at the LLOQ concentrations.

**Figure 6 molecules-24-00310-f006:**
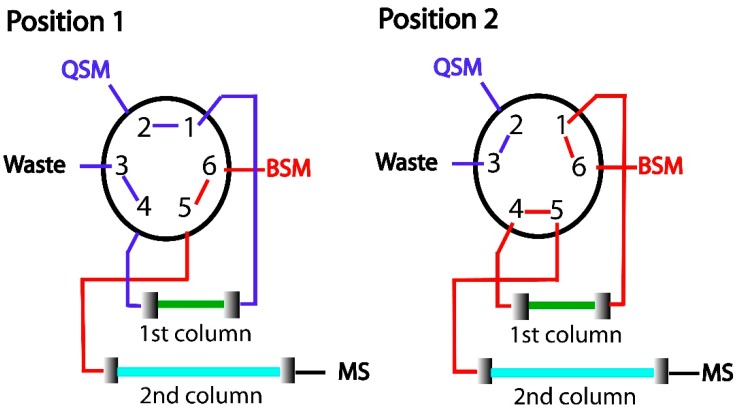
Scheme of in-tube SPME-UHPLC-MS/MS system. (**1**) Sample clean-up and sorption of the analytes, and (**2**) elution of the analytes from the 1st to 2nd column. QSM: quaternary pump, BSM: binary pump, MS: mass spectrometry, 1st column: monolithic capillary, 2nd column: analytical column.

**Table 1 molecules-24-00310-t001:** Optimization of the organic poly(butyl methacrylate-co-ethylene glycol dimethacrylate) monolith capillary synthesis procedure.

Monolith	Monomer/Porogen (%*m*/*m*)	EGDMA:BMA (%*m*/*m*)	Porogens BUT:PRO (%*m*/*m*)	AIBN	Permeability
M1	40:60	70:30	25:65	1%	Poor
M2	40:60	55:45	30:60	1%	Poor
M3	35:65	40:60	57:43	1%	Good
M4	35:65	50:50	57:43	1%	Good
M5	35:65	30:70	57:43	1%	Good

BUT = 1,4-butanediol; PRO = 1-propanol; AIBN = 2,2-azobisisobutylnitrile, BMA = butyl methacrylate, EGDMA = ethyleneglicol dimetacrylate.

**Table 2 molecules-24-00310-t002:** Ions transitions, instruments settings, and retention times for each antipsychotic and their metabolites.

Analyte	Precursor Ion (*m/z*)	Quantifier Ion (*m/z*)	Ce (eV)	DP (v)	Qualifier Ion (*m/z*)	Retention Time
Chlorpromazine	319.0	85.9	38	18	57.9	9.27
Chlorpromazine-d3	324.0	60.9	34	42	89.0	9.25
Clozapine	327.0	270.0	44	30	191.9	8.62
Olanzapine	313.0	256.0	22	20	84.0	5.95
Quetiapine	384.0	253.0	36	18	221.0	8.66
Quetiapine-d8	392.2	225.9	38	48	257.8	8.64
Desmethyl chlorpromazine	304.9	72.0	30	14	43.9	9.19
7-hydroxy-chlorpromazine	335.0	85.9	30	34	57.8	8.71
*N*-desmethyl clozapine	313.0	191.9	28	52	69.9	8.52
*N*-desmethyl olanzapine	299.0	197.9	26	28	212.9	5.46
Norquetiapine	296.0	209.9	54	26	138.9	8.61

**Table 3 molecules-24-00310-t003:** Linearity of the SPME-UHPLC-MS/MS method.

Analyte	Linearity		
	R^2^	Internal Standart	Lack of Fit Test
Chlorpromazine	0.9986	chlorpomazine-d3	0.848
Clozapine	0.9997	quetiapine-d8	0.226
Olanzapine	0.9989	quetiapine-d8	0.146
Quetiapine	0.9981	quetiapine-d8	0.888
Desmethyl chlorpromazine	0.9975	chlorpomazine-d3	0.420
7-hydroxy-chlorpromazine	0.9992	chlorpomazine-d3	0.166
*N*-desmethyl clozapine	0.9989	quetiapine-d8	0.196
*N*-desmethyl olanzapine	0.9997	quetiapine-d8	0.955
Norquetiapine	0.9985	quetiapine-d8	0.115

* *p*-value at a significance level of 0.05.

**Table 4 molecules-24-00310-t004:** Accuracy, precision, and matrix effects of the SPME-UHPLC-MS/MS method.

Analyte	Concentration (ng mL^−1^)	Accuracy		Precision		Matrix Effects (%CV)
		Intra-Assay (%RSE) *n* = 5	Inter-Assay	Intra-assay (%CV) *n* = 5	Inter-Assay	
Chlorpromazine	10	−19.4	−18.9	1.7	0.9	
	30	3.5	1.9	2.5	0.7	0.6
	200	−3.1	−10.9	4.8	7.8	
	300	−0.1	0.4	7.7	0.7	12.1
	400	−2.5	0.2	10.0	7.0	
Clozapine	10	14.7	14,7	14.2	3.3	
	30	14.6	6.7	9.1	12.1	3.6
	350	−1.0	−5.9	3.5	1.1	
	500	−2.5	−3.2	2.4	1.0	1.9
	700	1.0	−0.7	0.7	3.8	
Olanzapine	10	18.4	19.3	9.0	12.1	
	30	13.3	9.7	3.2	3.7	6.7
	100	−1.9	−4.2	1.2	8.3	
	150	5.7	6.2	4.8	3.3	7.9
	200	2.5	1.8	3.5	1.0	
Quetiapine	10	−18.8	−14.5	2.6	3.2	
	30	13.6	−10.5	7.7	1.6	1.5
	250	−9.5	−8.4	2.5	5.2	
	500	4.7	0.7	4.1	10.2	9.1
	600	−1.4	0.3	2.0	8.1	
Desmethyl chlorpromazine	10	−6.3	−9.5	1.7	4.5	
	30	−13.4	−13.5	1.6	2.7	9.9
	100	−2.2	−2.1	3.1	3.6	
	150	4.6	−1.8	2.4	6.8	10.6
	200	−7.9	−12.8	9.1	2.3	
7-hydroxy-chlorpromazine	10	−5.0	8.8	11.9	3.6	
	30	0,9	−4.4	1.5	4.3	13.5
	100	2.0	0.4	8.3	2.7	
	150	−3.9	5.5	3.0	6.6	14.3
	200	2.9	5.0	3.6	3.2	
*N*-desmethyl clozapine	10	7.5	10.4	0.8	2.3	
	30	−8.8	−0.1	9.1	7.3	2.1
	200	−4.3	−10.3	2.2	13.1	
	300	1.0	12.6	6.3	14.2	2.2
	500	−1.7	−2.5	1.3	10.0	
*N*-desmethyl olanzapine	10	11.0	15.6	0.7	6.0	
	25	−1.3	4.3	13.5	5.1	5.3
	100	−2.4	−3.6	9.1	3.0	
	150	0.8	12.8	10.1	3.6	6.1
	200	1.0	4.3	3.5	8.6	
Norquetiapine	10	19.9	18.4	4.1	3.4	
	30	7.5	7.5	9.5	9.2	3.1
	100	−4.6	−5.4	4.4	1.5	
	150	0.04	−6.1	6.7	11.7	5.3
	200	1.5	−1.4	1.0	7.2	

**Table 5 molecules-24-00310-t005:** Comparison of the in-tube SPME-UHPLC-MS/MS method with other counterparts to determine antipsychotics and their metabolites in biological samples.

Analytes	Matrix	Sample Amount (µL)	Chromatographic Separation (min)	Analytical Technique	LLOQ (ng mL^−1^)	References
Aripiprazole Olanzapine Paliperidone Ziprasidone	Plasma	200	8	Protein precipitation	10.0 (olanzapine)	Park et al. 2018 [27]
chlorpromazine, haloperidol, levomepromazine, olanzapine, risperidone, and sulpiride	Plasma	500	7	SPE (Oasis HLB)	13.2 (chlorpromazine) 2.9 (olanzapine)	Khelfi et al. 2018 [28]
Haloperidol, olanzapine, chlorpromazine, quetiapine, clozapine	Plasma	200	10	column switching LC-MS/MS (hybrid monolith with cyano groups)	0.075–0.188	Domingues et al. 2015 [8]
Clozapine, risperidone, and metabolites	Urine	500	10	MEPS (C18) UHPLC-PDA	100.0	Gonçalves et al. 2015 [26]
chlopromazine, clozapine, quetiapine, olanzapine and metabolites	Plasma	300	4.5	In-tube SPME-UHPLC-MS/MS	10.0	This work

**Table 6 molecules-24-00310-t006:** Concentrations of the target antipsychotic and their metabolites in plasma samples from Schizophrenic patients.

Drugs	Therapeutic Drug Monitoring Interval (ng mL^−1^)	Plasma Concentrations			
		Patient 1	Patient 2	Patient 3	Patient 4
Chlorpromazine	30–300	-	-	-	329
Clozapine	350–600	-	-	528	-
Olanzapine	20–80	85	-	-	-
Quetiapine	100–500	-	400	-	-
Desmethyl chlorpromazine	-	-	-	-	12
7-hydroxy-chlorpromazine	-	-	-	-	40
*N*-desmethyl clozapine	-	-	-	328	-
*N*-desmethyl olanzapine	-	19		-	-
Norquetiapine	-		61	-	-

**Table 7 molecules-24-00310-t007:** Chromatography conditions for In-tube SPME-UHPLC-MS/MS analysis.

QSM A = Water, B = Acetonitrile			BSM A = 10 mmol L^−1^ Ammonium acetate (0.1% Formic Acid) B = Acetonitrile		
T (min)	Pump	Flow Rate (µL min^−1^)	%A	Valve Position	Comments
0.0	QSM	100	100	1	Sample cleanup and drug pre-concentration into monolithic capillary
0.0	BSM	100	15	1	Analytical column conditioning
2.0	BSM	100	80	2	Analyte elution from the monolith column to the chromatographic column
5.50	BSM	300	80	1	Beginning of chromatography separation on the analytical column.
5.50	QSM	100	0	1	Cleanup of monolithic capillary
13.00	BSM	100	15	1	End of analytical separation and start of column conditioning for the next sample injection
13.0	QSM	300	100	1	Start of monolithic column conditioning for the next sample injection

QSM: quaternary pump, and BSM: binary pump.

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
