# Peer review of "Butyl Methacrylate-Co-Ethylene Glycol Dimethacrylate Monolith for Online in-Tube SPME-UHPLC-MS/MS to Determine Chlopromazine, Clozapine, Quetiapine, Olanzapine, and Their Metabolites in Plasma Samples"

_molecules, 2019, doi:10.3390/molecules24020310_

Round 1
Reviewer 1 Report
Butyl methacrylate-co-ethylene glycol dimethacrylate monolith for online in-tube SPME-UHPLC-MS/MS to determine chlopromazine, clozapine, quetiapine, olanzapine, and their metabolites in plasma samples
Authors: Luiz G. M. Beloti, Luis F. C. Miranda and Maria Eugênia C. Queiroz
This manuscript describes the development (including the synthesis of the SPME coating) of an analytical methodology based on in-tube SPME-UHPLC-MS/MS to determine 4 drugs and their metabolites in plasma samples.
The manuscript offers novelty and can be interesting for readers. However, I have several minor comments for authors.
- The organization of the manuscript results confused. Please move the Section: Results and Discussion (from page 2, line 72 to page 9, line 174), after the Section: Materials and Methods.
- Page 5, line 130. Although to select the SRM transitions, Full Scan mode is employed in a first step, the term full scan can be confused for readers, since the LC-MS/MS analysis was finally performed under SRM mode.
- Table 3. Since the determination coefficient (R2) is provided, the linear equation does not offer additional information, therefore it should be removed from the Table
- Table 5. Although in page 8, line 159 it is compared the LOQ of the proposed method with those reported in the literature, these values should be also included in Table 5 to show a complete comparison.
- Please unify the term SRM or MRM through the manuscript
Author Response
Dear Editor,
The authors would like to thank the reviewers for all the recommendations. The manuscript has been revised in attendance to the statements made by the reviewers.
Review 1
This manuscript describes the development (including the synthesis of the SPME coating) of an analytical methodology based on in-tube SPME-UHPLC-MS/MS to determine four drugs and their metabolites in plasma samples.
The manuscript offers novelty and can be interesting for readers. However, I have several minor comments for authors.
- The organization of the manuscript results confused. Please move the Section: Results and Discussion (from page 2, line 72 to page 9, line 174), after the Section: Materials and Methods.
The manuscript was organized in agreement with the template supplied by Molecules Journal. The template is available at https://www.mdpi.com/journal/molecules/instructions website.
- Page 5, line 130. Although to select the SRM transitions, Full Scan mode is employed in a first step, the term full scan can be confused for readers, since the LC-MS/MS analysis was finally performed under SRM mode.
This sentence has been modified in the manuscript (page 3, line 136).
- Table 3. Since the determination coefficient (R2) is provided, the linear equation does not offer additional information, therefore it should be removed from the Table.
The linear equation of all the analytes has been removed from Table 3.
- Table 5. Although in page 8, line 159 it is compared the LOQ of the proposed method with those reported in the literature, these values should be also included in Table 5 to show a complete comparison.
The LLOQ values have been included in Table 5.
- Please unify the term SRM or MRM through the manuscript
The corrections have been made.

Reviewer 2 Report
Molecules
Review of Manuscript Number: molecules-421522
TITLE: Butyl methacrylate-co-ethylene glycol dimethacrylate monolith for online in-tube SPME-UHPLC-MS/MS to determine chlopromazine, clozapine, quetiapine, olanzapine, and their metabolites in plasma samples.
CORRESPONDING AUTHOR: Dr. Maria Queiroz
General Comments
This manuscript addresses the development and evaluation of determination of several antipsychotics for simple sample preparation approach. The prepared monolith capillary was applied to actual biological samples with sensitivity, the possibility of application was expected for actual samples in the future. Although, I found some of the authors' explanations difficult to follow; I suspect a reader less familiar with the topic might have even greater difficulties. I would suggest major revision in combination with re-review for this manuscript. I annotate the manuscript with several corrections listed below, which I believe will improve the readability of the paper.
1: Major comments
(1) The authors discussed a property of synthesized monolith capillary for determination of antipsychotics, herein, the analytical validation was well carried out for the optimization of LC-MS condition. On the other hand, however, novelty of the synthesized organic monolith and comparison with other monolith materials were not discussed sufficiently in main text. Please insert the explanations of novelty of this study in the manuscript (or clarified the position of this study in therapeutic drug monitoring analytical field).
(2) The authors used the prepared organic monolith capillary as pre-concentrated column; the sensitive determination of antipsychotics was achieved. The authors explained the process as hydrophobic interactions between analytes and the capillary (Page 5, Line 127). However, the pre-concentrated mechanism of antipsychotics to monolithic capillary was not sufficiency discussed in manuscript, because the simple hydrophobic interaction has relatively low selectivity. It is seemed that the monolith capillary was used as a simple solid-phase column, hence I suggest that the addition of description for selectivity of antipsychotics from other drugs in Results and Discussion section. (In other words, can the monolith capillary apply to detect other therapeutic drugs?)
Alternatively, the article could be moved into a short communication waiting to test other substances.
(3) The authors applied organic monolith capillary for TDM of schizophrenic patients in Table 6, however, the discussion was not sufficiency in main text. Please insert the discussion from the results such as including metabolic process of antipsychotics.
In addition, norquetiapine was detected from plasma levels 1, and N-desmethyl olanzapine was detected from plasma levels 2. These metabolic processes should be explained in main text.
2: Minor comments
(1) There are a lot of paragraphs in Introduction section. The section should be summarized with simple paragraphs.
(2) The chemical structures of the antipsychotics should be inserted in manuscript. The chemical structure information will provide an assist of consideration of metabolic process in biological samples.
(3) Page 2, line 57
The reference numbers (Ref. 15-17) are not serialization. Please check the manuscript and references form again.
(4) Figure 4
Please insert the TIC in Figure 4. The information will provide the discussion of sensitivity from biological samples.
(5) Page 10, Line 189
Please confirm the unit of resistance of Milli-Q water. (18 “M”ohm?)
I hope that my comment is very useful for the improvement of the article.
Author Response
Dear Editor,
The authors would like to thank the reviewers for all the recommendations. The manuscript has been revised in attendance to the statements made by the reviewers.
The responses are in PDF file..

Round 2
Reviewer 1 Report
The manuscript has been improved, and therefore it can be published in the journal in the present form.
Reviewer 2 Report
Molecules
Review of Manuscript Number: molecules-421522
TITLE: Butyl methacrylate-co-ethylene glycol dimethacrylate monolith for online in-tube SPME-UHPLC-MS/MS to determine chlopromazine, clozapine, quetiapine, olanzapine, and their metabolites in plasma samples.
CORRESPONDING AUTHOR: Dr. Maria Queiroz
General Comments
The authors have substantially revised their paper and thus improved its quality. The authors’ replies to reviewer comments are detailed and convincing and show that they have taken seriously the reviewers’ advice.
Minor comment
Please write the m/z as italics in Table 2.